# The Clinical Impact of MicroRNA-21 in Low Rectal Cancer Treated with High-Dose Chemoradiotherapy in the Organ Preserving Setting

**Caroline Brenner Thomsen** [1,*] , **Rikke Fredslund Andersen** [1,2] , **Lars Henrik Jensen** [1,3] , **Anders Jakobsen** [1,3] **and Torben Frøstrup Hansen** [1,3]

1    Danish Colorectal Cancer Center South, Vejle Hospital, University Hospital of Southern Denmark, 7100 Vejle, Denmark; Rikke.Fredslund.Andersen@rsyd.dk (R.F.A.); Lars.Henrik.Jensen@rsyd.dk (L.H.J.); Anders.Jakobsen@rsyd.dk (A.J.); Torben.Hansen@rsyd.dk (T.F.H.)

2    Department of Clinical Biochemistry, Vejle Hospital, University Hospital of Southern Denmark, 7100 Vejle, Denmark

3    Institute of Regional Health Research, University of Southern Denmark, 5000 Odense, Denmark

*    Correspondence: Caroline.Emilie.Brenner.Thomsen@rsyd.dk; Tel.: +45-79406038

**Abstract: Background:** Organ preservation in the treatment of rectal cancer has seen an increase in interest. Clinical complete response (cCR) after high-dose chemoradiotherapy (CRT) allows for non-surgical management (NSM), but the selection of patients is challenging and standard clinical staging insufficient. MicroRNA-21-5p (miR-21) is ubiquitously upregulated in cancer and has been associated with treatment response in rectal cancer treated with standard preoperative CRT. The aim of the present study was to investigate this association in low rectal cancer treated in the NSM setting. **Methods:** Forty-eight patients from our single-arm phase II trial (NCT00952926) were eligible for analysis. All patients had resectable T2 or T3, N0–N1 low adenocarcinoma and received intensity-modulated radiotherapy plus brachytherapy boost and oral tegafur–uracil. Patients with cCR six weeks after end of treatment assessed by clinical examination, magnetic resonance imaging, and biopsy, were referred to observation and close follow-up. The miR expression in the diagnostic biopsies was measured by qPCR. The relationship between miR-21 expression and cCR was assessed using the Wilcoxon rank-sum test. **Results:** Thirty-eight patients had cCR after treatment and were allocated to observation while 10 patients had incomplete response and underwent surgery. MicroRNA-21 was successfully analyzed in all samples. The median tumor expression of miR-21 was significantly higher in patients with incomplete response than in those with cCR, 24.3 (95% confidence interval (CI) 17.1–36.8) and 16.6 (95% CI 13.9–21.1), respectively, *p* = 0.03. **Conclusions:** The present study adds to the evidence of the clinical impact of miR-21 in rectal cancer treated with CRT. The findings are comparable with results seen in patients treated in the standard preoperative setting and may assist in the selection of patients for an organ preserving approach.

**Keywords:** rectal cancer; microRNA 21; biomarker

## 1. Introduction

Surgery is the main treatment modality in localized rectal cancer, but it has potentially disabling side effects. The postoperative mortality is around 2% with a much higher frequency of postsurgical morbidity [1]. Some of the complications are long-term such as pain, faecal incontinence, and malfunction of the bowel. Furthermore, there may be a patient preference for avoiding stoma. Small rectal tumors are managed with local resection whereas patients with larger tumors undergo extensive surgery and represent a particular, therapeutic challenge.

In recent years, high-dose preoperative chemoradiotherapy (CRT) has proven to reduce local recurrence and often results in significant tumor regression, sometimes even with clinical complete response (cCR) [2–4]. However, the reduction in tumor load after preoperative CRT varies considerably. Studies have shown that up to 40% of patients may be eligible for an organ preserving approach following high-dose CRT [5] without compromising local disease control or survival and with a higher quality of life. Identification of molecular markers capable of predicting treatment response would allow for a more accurate selection of candidates for observation after curatively intended high-dose CRT.

MicroRNAs (miRs) are short, single stranded, non-coding RNA molecules functioning as gene regulators at the posttranscriptional level. They play an important role in the regulation of biological processes such as cell proliferation, differentiation, and apoptosis, and clinical studies have shown an association between treatment response and miR profiles. MicroRNA-21 is one of the most investigated miRs in colorectal cancer, as it is consistently upregulated compared to normal tissue and found to be important in tumor progression and metastasis [6,7]. In a recent meta-analysis, miR-21 demonstrated promising abilities as a prognostic biomarker in rectal cancer [8]. Furthermore, the level of miR-21 appears to be predictive of response to high-dose CRT [9]. The present study investigated the role of miR-21-5p (miR-21) in treatment-naive rectal tumor biopsies to help select patients for observation after curatively intended high-dose CRT.

## 2. Results

Of the 48 patients included in the study, 38 (79%) achieved cCR six weeks after high-dose CRT and were allocated to the observation group. Ten patients (21%) had incomplete response and were referred to surgery. MicroRNA-21 was successfully analyzed in all samples. Patient characteristics at baseline (N = 48), according to the median level of miR-21 in the observation group (lower or higher than the median level) (N = 19 in both groups), overall in the observation group (N = 38), and in the incomplete response group (N = 10) are shown in Table 1.

**Table 1.** Patient characteristics at baseline (all patients), divided by the median level of MicroRNA-21-5p (miR-21) in the observation group (median miR-21 = 16.6), in the overall observation group, and the group of patients with incomplete response.

| | All Patients (N = 48) | Patients with miR-21 *below* the Median Level (N = 19) | Patients with miR-21 *higher* Than the Median Level (N = 19) | Observation (N = 38) | Incomplete Response (N = 10) |
|---|---|---|---|---|---|
| Parameter | N (%) | N (%) | N (%) | N (%) | N (%) |
| **Age, years** | | | | | |
| Median (range) | 67 (32–86) | 72 (46–83) | 68 (50–86) | 68 (46–86) | 54 (32–69) |
| **Gender** | | | | | |
| Female | 10 (20) | 4 (21) | 3 (16) | 7 (18) | 3 (30) |
| Male | 38 (80) | 15 (79) | 16 (84) | 31 (82) | 7 (70) |
| **PS** | | | | | |
| 0–1 | 46 (96) | 17 (89) | 16 (84) | 37 (97) | 9 (90) |
| 2 | 1 (2) | 2 (11) | 2 (11) | 0 | 1 (10) |
| ND | 1 (2) | 0 | 1 (5) | 1 (3) | 0 |
| **Tumor stage** | | | | | |
| T2 | 26 (54) | 10 (53) | 12 (63) | 22 (58) | 4 (40) |
| T3 | 22 (46) | 9 (47) | 7 (37) | 16 (42) | 6 (60) |

As appears, there was no difference in patient characteristics according to the median miR-21 level. The median age was lower and clinical tumor stage higher in patients with incomplete response compared to patients with cCR.

In the observation group, the median tumor expression of miR-21 was 16.6 (95% CI 13.9–21.1). In patients without cCR, the median level was 24.3 (95% confidence interval (CI) 17.1–36.8). The difference was statistically significant ($p = 0.03$), Figure 1A,B.

(**A**) The median is illustrated by the black horizontal line in the box. The whisker boundaries are the minimum and maximum level of miR-21. The box edges represent the 50% inter-quartile ranges (*p*-value 0.03).

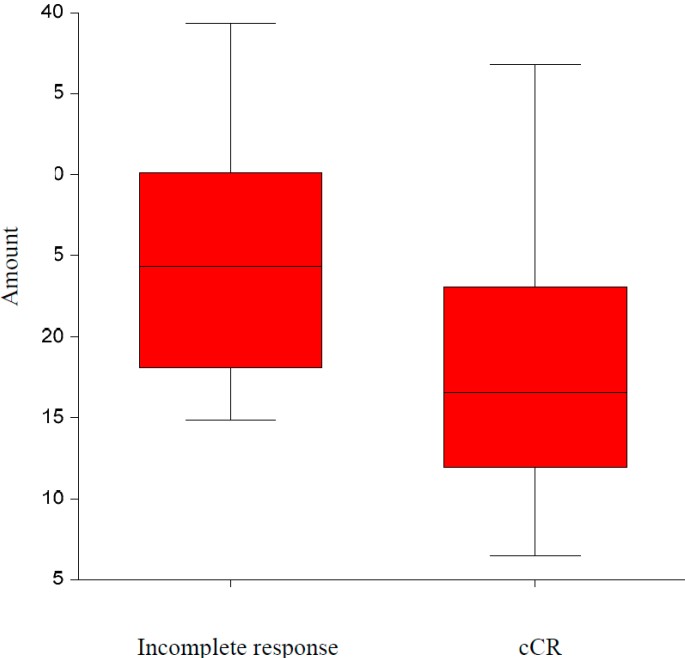

(**B**) Dot plot showing the same data with all patients represented by a dot.

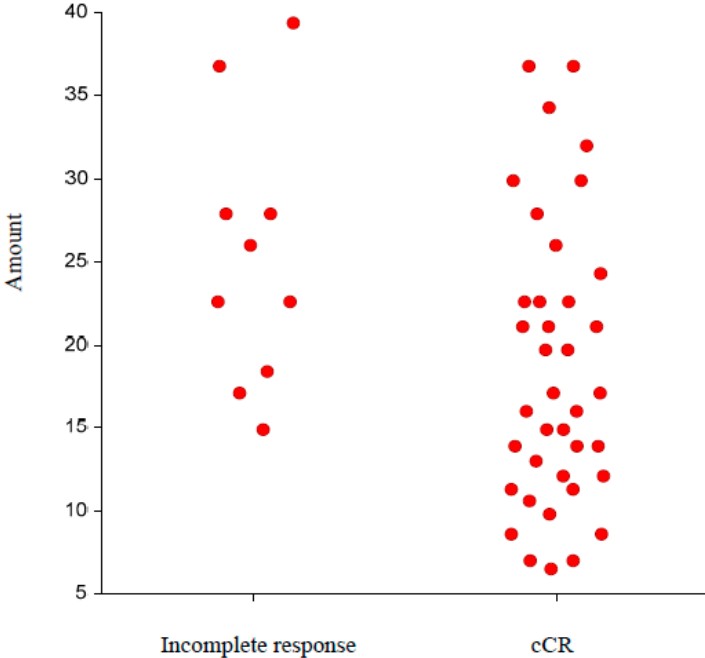

**Figure 1.** (**A**,**B**) The level of miR-21 in patients with (N = 38) and without (N = 10) clinical complete response (cCR) illustrated in a box and whisker plot (**A**), and dot plot (**B**).

The positive and negative predictive values of undergoing surgery based on the median tumor expression of miR-21 were 95% (19/20) and 32% (9/28), respectively.

## 3. Discussion

This prospective observational study with retrospective miR-21 analysis of rectal cancer patients identified a potential role of miR-21 in the selection of patients for organ preservation.

One of the main challenges of rectal cancer management is the selection of patients for observation and hence omission of mutilating surgery. So far, there are no clinically validated biomarkers, but miR-21 has previously been investigated as a potential predictive marker of cCR. A study by Lopes-Ramos et al. demonstrated a higher expression of miR-21 in rectal cancer patients with cCR after neoadjuvant CRT than detected in incomplete responders [10]. This is in contrast to our results, but their study varied from ours in several central aspects. First of all, they included more advanced cancers with larger tumors (T4) and N2 disease. The radiotherapy dose was lower and did not include brachytherapy. Additionally, they evaluated treatment effect at least 12 weeks after end of treatment compared to 6 weeks in our study. Previous results from our group, based on two cohorts of patients treated with high-dose CRT, demonstrated a trend towards better outcome for tumors with high miR-21 expression [11]. This is not directly comparable with the present results, however, since treatment dosing and tumor sizes were not the same. On the other hand, a recent study by Caramés et al. reported results equal to those presented here [9]. They evaluated 70 rectal cancer patients and found an odds ratio of overexpressed miR-21 and non-cCR of 9.75 (95% CI 2.24–42) confirmed in multivariate analysis. The number of patients with cCR was remarkably lower than ours, 14% (10/70) compared to 79% (38/48). The main difference between the studies is the design. Caramés et al. performed a retrospective study of patients who all underwent surgery. Furthermore, they defined a cut-off value of miR-21 at 2.8, which is difficult to validate. Another study published in 2018 confirmed the predictive role of miR-21 in rectal cancer, although the combination of four miRs had a better predictive capacity [12].

Our study has limitations. It is a small, retrospective study without a validation cohort, and the analysis of miR-21 was not pre-planned.

The high, positive predictive value of miR-21 in this study is promising. Validation of the present findings is planned in a recently completed and comparable cohort after sufficient follow-up (ClinicalTrials.gov identifier: NCT02438839).

Analysis of miR-21 is low-cost and easy to perform, but lack of standardization in sampling, analysis, and interpretation is an obvious challenge, which needs further attention before miRs may be implemented as clinical decision tools. MiR-21 is almost ubiquitously upregulated in cancers and the clinical impact of this essential miRNA, and the numerous downstream targets affected, has been reviewed several times [8,13] underlining its potential as biomarker.

## 4. Materials and Methods

### 4.1. Study Population

Fifty-one patients with biopsy-verified low rectal adenocarcinoma and primary resectable clinical T2 or T3 tumor were enrolled in a prospective, open, observational study at Danish Colorectal Cancer Center South, Vejle Hospital, Denmark, between July 2009 and July 2012, as previously published [14]. Since in three cases there was not sufficient tumor tissue for miR-analysis, 48 patients were included in the present retrospective biomarker study. Additional inclusion criteria were planned; abdominoperineal or ultra-low resection, distance from anal verge to lower margin of tumor less than 6 cm, suited for curative intent radiation and chemotherapy, consent to biopsy and blood samples for translational research, and age above 18 years. Written and orally informed consent was provided by all participants. The study was approved the 16th of June 2009 by The Regional Committees on Health Research Ethics for Southern Denmark (S-20090063) and registered with clinicaltrials.gov (NCT00952926).

### 4.2. Treatment Course and Evaluation

The patients received intensity-modulated radiotherapy in 30 fractions with 60 Gy to the primary tumor clinical target volume, 50 Gy to the elective clinical target volume, and a single brachytherapy boost of 5 Gy to the tumor. Concomitant oral tegafur–uracil 300 mg/m$^2$ was administered on treatment days in weekly cycles of five days.

Treatment effect was evaluated by clinical examination including endoscopy and digital rectal exam, magnetic resonance imaging (MRI) of the pelvis, computed tomography (CT) scan of the chest and abdomen, and biopsy at baseline and week two, four, and six during treatment.

Patients with cCR were offered observation based on no residual tumor at clinical examination at week 6, negative biopsies, no lymph node metastases on MRI (no malignancy signs and lymph nodes less than 3 mm), and no metastases on the CT scan. Patients not fulfilling these criteria were referred to surgery 8 weeks after start of treatment. Further details are available from our previous publication [14].

### 4.3. MicroRNA Analysis

MicroRNA analysis was performed on the diagnostic biopsies. If tumor tissue was sparse, laser microdissection was performed (71%). The methods are described in detail in previous publications from our group [15,16]. In brief, RNA was isolated using miRNeasy FFPE Kit (Qiagen, Hilden, Germany) according to the manufacturer's instructions. The miR expression was measured by qPCR in triplicates on a QuantStudio 12K Flex real-time PCR system in 20 μL reactions using TaqMan MicroRNA Assays (Thermo Fisher, Waltham, MA, USA). An example of a qPCR plot is shown in Supplementary Figure S1. The protocol using custom RT and preamplification pools was followed (User guide 4465407, Thermo Fisher). miR-193a-5p, -27a and -let7g were used for normalization [15].

### 4.4. Follow-Up

The follow-up program for patients in the observation group consisted of clinical examination every two months the first year, every three months the second year, every six months the third year, and annually the fourth and fifth year. The clinical controls were supplemented by positron emissions tomography (PET)-CT scans 6, 12, 18, 24, 36, 48, and 60 months after end of treatment.

Patients with incomplete response were followed according to the guidelines of the referring department of surgery.

Quality of life was assessed by means of patient completed questionnaires, and the long-term results were published recently [17].

### 4.5. Statistical Analysis

The relationship between miR-21 expression and cCR was assessed using the Wilcoxon rank-sum test. Reported *p* values were two-sided and $p < 0.05$ was considered statistically significant. Statistical tests were performed using the NCSS 10 Statistical Software (2015) (NCSS, LLC. Kaysville, Utah, USA, ncss.com/software/ncss).

## 5. Conclusions

In conclusion, our study supports a clinical impact of miR-21 in localized rectal cancer treated with high-dose CRT. The level of expression may predict treatment response and be instrumental in the selection of patients for organ preservation.

**Supplementary Materials:** Supplementary materials can be found at http://www.mdpi.com/2624-5647/2/4/34/s1.

**Author Contributions:** Conceptualization, A.J., L.H.J. and T.F.H.; methodology, R.F.A.; formal analysis, C.B.T. and T.F.H.; investigation, T.F.H. and C.B.T.; writing—original draft preparation, C.B.T.; writing—review & editing, all authors. Supervision, T.F.H. All authors have read and agreed to the published version of the manuscript.

**Funding:** This research received financial support from The Regional Strategic Council for Research in the Region of Southern Denmark.

**Acknowledgments:** We would like to thank the study participants and the staff at Vejle Hospital who collected samples for this study. Additionally, we gratefully acknowledge the technicians Pia Nielsen, Tina Brandt Christensen, and Lone Hartmann Hansen. Finally, we express gratitude to Karin Larsen for linguistic editing of the manuscript.

**Conflicts of Interest:** The authors declare no conflict of interest.

## Abbreviations

cCR Clinical complete response
CRT Chemoradiotherapy
miR-21 MicroRNA-21-5p
miRs MicroRNAs
MRI Magnetic Resonance Imaging
CT Computed Tomography
PET-CT Positron-Emissions-Tomography CT
CI Confidence interval

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
