# Peer review of "The Clinical Impact of MicroRNA-21 in Low Rectal Cancer Treated with High-Dose Chemoradiotherapy in the Organ Preserving Setting"

_gastrointestdisord, doi:10.3390/gidisord2040034_

Round 1
Reviewer 1 Report
Even though it is a brief observational report, the authors intend to implicate the clinical significance of MicroRNA-21 in rectal cancer. To that end, they analyze MicroRNA-21 expression in a small population in post-operative setting and correlate it with clinical outcome. The study however, is not validated in any way. Neither do they have a validation group, nor they have any investigational data using animal models or in vitro studies. The study needs to be strengthened at least with some in silico retrospective study or metanalysis of some kind.
Author Response
- Thank you for the comments. The aim of this study is solely clinical. Can we, by this test, help selecting patients for an organ preserving strategy already at the time of diagnosis? There are already, as sited in this paper, sufficient clinical data to support this hypothesis. With that perspective, additional data from animal, or in vitro, studies is not necessary. This strategy, in general, is still experimental (not standard) and there are no randomized studies at all, so a direct validation cohort does not exist. We are aware of these limitations as specified in the Discussion section (page 6, line 143-144). Because of these, we wrote this brief report with the hope that other groups may be able to validate these results in as proper a manner as possible.
Reviewer 2 Report
This is descriptive short report on miRNA in colorectal patients. The study can be improved by presenting RBA analysis, show representative RNA isolated samples QPCR plots. The Figure 1 legend need to add A. Need to add p-value to Figure 1. Table 1, add p-values. In Discussion, please add signaling downstream of the described MiRNA, analyze gene expression database data on regulated signaling.
Author Response
This is descriptive short report on miRNA in colorectal patients.
The study can be improved by presenting RBA analysis, show representative RNA isolated samples QPCR plots.
We thank the reviewer for the comment. We have now added an example of a qPCR plot as Supplementary Figure 1.
The Figure 1 legend need to add A. Need to add p-value to Figure 1. Table 1, add p-values.
We are not sure what is meant by “The Figure 1 legend need to add A“. The ‘A’ and ‘B’ are now highlighted. The p-values have been added in Figure 1. In our opinion, adding p-values in Table 1 does not add value to the table. The distribution of patients are alike when compared in below and higher than the median miR-21 level. In the groups of ‘Observation’ and ‘Incomplete response’, there are only 10 patients in the later group, and by looking at the numbers, all other demographics than age are similar distributed.
In Discussion, please add signaling downstream of the described MiRNA, analyze gene expression database data on regulated signaling.
We thank the reviewer for the notification and recognize the importance of the signaling downstream from mRNA-21.. In the Discussion section, we have now added a part on the importance of miR-21 (page 6).
Reviewer 3 Report
This is a useful and clinically highly relevant paper correlating expression of a miRNA with clinical response to non surgical treatment of rectal cancer. Treatment of this disorder has advanced over the years, with sphincter-preserving surgery being the norm except on those cases in which it is not possible. Whole organ preserving treatment may be possible with proper methodology, but it is important to stratify patient candidates to groups that would benefit from this approach. This paper leads in this direction, the methodology, data, and conclusions are sound and the limitations and future directions are properly indicated. I therefore recommend acceptance.
Author Response
We thank the reviewer for the comments.
Round 2
Reviewer 1 Report
As mentioned in first round of review, the study does not seem adequate to me. The response is not convincing and in my opinion a little more detail is required even for a short report.
Author Response
Thank you. We appreciate the observations and value the criticism in which we highly agree. In our opinion, the preclinical studies have already been performed, as stated in the manuscript and in our first reply. A number of publications have highlighted the potential clinical impact of miR-21 in patients with rectal cancer undergoing pre-operative radio-chemotherapy. We thus find it appropriate, and important, to precede to this specific clinical setting where the treatment is very much similar but with the aim of avoiding surgery (organ preserving). Hence, we do not see the need for pre-clinical testing in this specific scenario as clinical evidence is already present. We believe this is sufficient for the testing of our hypothesis.
We support the reviewers comment on the need of validation. The validation of clinically studied biomarkers is essential. Our cohort is small, as this treatment approach is still experimental and the results obviously only serve as hypothesis generating. We do not indicate that our results (for now) play a role in the standard therapy of rectal cancer. We hope that our results will be validated by other research centers in similar patient cohorts, and we are planning to evaluate it in a currently ongoing protocol addressing the same treatment approach.
As indicated, we highly agree to the comments, but for the reasons stated above preclinical testing is not called for and validation will be performed, in time, but cannot be accommodated now.